# Molecular Mechanisms of the Antitumor Effects of Mesalazine and Its Preventive Potential in Colorectal Cancer

**DOI:** 10.3390/molecules28135081

**Published:** 2023-06-29

**Authors:** Joanna Słoka, Marcel Madej, Barbara Strzalka-Mrozik

**Affiliations:** Department of Molecular Biology, Faculty of Pharmaceutical Sciences in Sosnowiec, Medical University of Silesia, 40-055 Katowice, Poland; mmarcel281297@gmail.com (M.M.); bstrzalka@sum.edu.pl (B.S.-M.)

**Keywords:** mesalazine, colorectal cancer, Wnt/β-catenin pathway, chemoprevention, cancer stem cells, non-steroidal anti-inflammatory drugs, molecular mechanism, inflammation

## Abstract

Chemoprevention is one of the ways to fight colorectal cancer, which is a huge challenge in oncology. Numerous pieces of evidence indicate that chronic inflammation in the course of Crohn’s disease or ulcerative colitis (UC) is a significant cancer risk factor. Epidemiologic studies suggest that long-term use of non-steroidal anti-inflammatory drugs (NSAIDs), including mesalazine, has beneficial effects on colitis-associated colorectal cancer. Mesalazine is a first-line therapy for UC and is also widely used for maintaining remission in UC. Data showed that mesalazine has antiproliferative properties associated with cyclooxygenase (COX) inhibition but can also act through COX-independent pathways. This review summarizes knowledge about mesalazine’s molecular mechanisms of action and chemopreventive effect by which it could interfere with colorectal cancer cell proliferation and survival.

## 1. Introduction

Colorectal cancer (CRC) is one of the most significant healthcare problems worldwide. It is the second most commonly diagnosed cancer in women and the third in men [1,2]. In terms of mortality, it is the second leading cause of cancer-related death in the world [3]. Approximately 25–30% of confirmed colorectal cancers lead to metastasis, and nearly 50% of colorectal cancer patients have recurrence [4,5]. The mainstay of CRC treatment is tumor resection and chemotherapy. Additionally, radiotherapy is used before or after surgery, and in the case of disseminated CRC, targeted therapy such as anti-epidermal growth factor receptor (anti-EGFR) antibodies and anti-vascular endothelial growth factor receptor antibodies (anti-VEGFR) is used [6,7].

There are many risk factors leading to CRC development. We can distinguish between colorectal cancer risk factors depending on lifestyles and unmodified ones regardless of health concerns. An unhealthy lifestyle plays a major role in causing colorectal cancer. Environmental factors, such as overuse of alcohol, a high-calorie diet, lack of exercise, and carcinogens derived from frying foods, play a significant role in the development of CRC [6,7,8,9]. Furthermore, smokers have a 48% higher risk of developing CRC compared to nonsmokers [10]. Conversely, consumption of whole grains, fresh fruits and vegetables, and regular physical activity can decrease the risk. On the other hand, there are many well-known risk factors related to developing colorectal cancer independent of lifestyle. These include hereditary syndromes such as Lynch syndrome and familial polyposis associated with the APC gene mutation [6,11,12].

Inflammatory bowel disease (IBD) significantly increased the risk of CRC developing because of chronic intestinal inflammation, which leads to tumor development [13,14]. Inhibition of inflammatory processes correlates with decreased tumor formation. Numerous studies have indicated that anti-inflammatory drugs cause inhibition of tumor growth and angiogenesis and also increase apoptosis [15,16,17]. Furthermore, Cho et al. [1] show that a healthy lifestyle is associated with a reduced risk of colorectal cancer, regardless of an individual’s genetic risk factors. Due to this information, there has been a recent increase in the amount of ongoing research into the potential cytotoxic and antiproliferative effects of various therapeutic substances, among which mesalazine, an anti-inflammatory drug, can be included. Recent studies indicate that the mechanism of action of mesalazine can also be used in the treatment and prevention of colorectal cancer. This is due to its comprehensive action on the pathways responsible for the development and progression of cancer, i.e., the Wnt/β-catenin pathway and PPARγ, as well as its effect on the cell cycle or inhibition of the proliferation of cancer stem cells, which are the main cause of CRC recurrence. Taken together, the present paper aims to review the molecular basis of mesalazine’s chemopreventive effect in colorectal cancer, which may imply new strategies in the struggle with colorectal cancer. Mesalazine offers the possibility of chemoprevention at the stage of tumor formation and may also contribute to the prevention of its recurrence.

## 2. Mesalazine—An Old Drug, New Possibilities

The new idea of “drug repurposing”, involves attempts to use known and frequently used drugs for other indications. The main advantage of this approach over the synthesis of *de novo* drugs is a significant reduction in the costs and time required to implement a new therapy. In addition, a drug that has been available in medicine for a long time is well-established in terms of pharmacotherapy safety, pharmacokinetics, and interactions with other drugs [5,18]. One example with potential use in the treatment of colorectal cancer is mesalazine. Mesalazine, also known as mesalamine or 5-aminosalicylic acid (5-ASA), belongs to the non-steroidal anti-inflammatory drugs (NSAIDs) commonly used to treat inflammatory bowel disease (IBD). It is also used in the chemoprophylaxis of CRC associated with these conditions. Available data indicate that mesalazine also has the potential to inhibit tumor cell proliferation through several pathways [19]. As a medicinal substance, mesalazine is the pharmacologically active part of sulfasalazine, the first compound used to treat ulcerative colitis. Sulfapyridine, on the other hand, is the inactive component of sulfasalazine and is responsible for severe side effects. Consequently, mesalazine is preferred for self-administration [20,21].

Mesalazine is available in several forms, mainly as tablets, suppositories, foams, or as an enema [22]. 5-ASA can also be combined with sulfapyridine, as mentioned earlier, known as sulfasalazine. Two 5-ASA molecules joined by an azo bond are known as olsalazine. Application of 5-ASA in a conjugate form significantly reduces the absorption of the drug in the small intestine, allowing for sufficiently high concentrations in the terminal ileum and colon. The conjugates are cleaved in the colon by azoreductase. About 20–30% of 5-ASA from mesalazine preparations is absorbed in the small intestine and N-acetylated in intestinal epithelial cells and the liver. About 25% of mesalazine is absorbed in the large intestine, and the remainder is excreted unchanged in the feces [23,24]. 5-ASA, compared with other NSAIDs, is relatively well tolerated and has few side effects after use [25,26]. Rarely, interstitial nephritis, alveolitis, and pancreatitis have been observed; however, these side effects are usually reversible after discontinuation of mesalazine therapy [25,26].

IBD includes two main phenotypes: ulcerative colitis (UC) and Crohn’s disease (CD) [27]. Mesalazine is a first-line therapy in UC, resulting in good treatment outcomes in 88% of patients [28]. In addition, it is widely used to maintain remission in UC [29]. The mechanism of 5-ASA’s anti-inflammatory effect is not fully explained; however, existing data indicate that 5-ASA antagonizes pro-inflammatory mediators such as interferon-γ, IL-8, nuclear factor-κB, and tumor necrosis factor-α [29,30,31,32,33]. Mesalazine also inhibits the cyclooxygenase (COX) and lipoxygenase (LOX) pathways, contributing to the inhibition of prostaglandin E2 and leukotriene release [34], which are strongly associated with inflammation. It is also believed that the increase in PPARγ expression in gastrointestinal epithelial cells by mesalazine may be another mechanism of anti-inflammatory action [28]. Mesalazine also has an antioxidant function and is a free radical scavenger [35]. The aforementioned mechanisms result in the conclusion that, as a therapeutic substance, it may prevent intestinal inflammation but also induce mucosal healing [36]. As a compound, it has been extensively studied in vitro [37,38,39]. Numerous clinical trials with mesalazine are also conducted, e.g., in the chemoprevention of diseases related to the large intestine. Examples of such studies are shown in Table 1, generated from the https://clinicaltrials.gov database (accessed on 1 April 2023).

## 3. Chronic Inflammation Triggers CRC Development

The first suggestion that inflammation might be associated with cancer development comes from the nineteenth-century physician Rudolf Virchow [40]. Since then, numerous data confirmed that chronic inflammation may stimulate tumor initiation, promotion, and progression so-called “cancer-promoting inflammation’’ [41]. IBDs are chronic inflammatory illnesses that are commonly associated with a higher risk of colorectal cancer development. Although the exact reason for IBD is still elusive, it is generally accepted that IBD is associated with external environmental factors, immunological abnormalities, and genetic susceptibility of the host or intestinal microbiota [42]. Long-term inflammation may lead to colitis-associated cancer (CAC), wherein the severity and time of active disease correlate with the risk of CAC development. Chronic inflammation of IBD patients reveals a higher level of pro-inflammatory cytokines. Especially levels of IL-1, IL-6, IL-8, IL-12, and tumor necrosis factor α (TNF-α) are increased in inflamed mucosa [43,44]. Intestinal cytokine network disruption might trigger colorectal cancer development. For instance, IL-6 is a pro-inflammatory cytokine that exerts a pro-tumorigenic effect by activating the Janus kinases (JAK) and signal transducers and transcription activators (STATs), which are strongly associated with tumorigenic processes [17,45]. Interestingly, chronic inflammation and cancer development work on positive feedback mechanisms. Inflammation predisposes cancer development, but newly emerging tumors sustain an inflammatory process that stimulates its progression. During chronic inflammation, different inflammatory mediators are released, which activate transcriptional factors [15]. This is well described by two pathways connecting inflammation and cancer: an extrinsic and an intrinsic pathway. The extrinsic pathway is related to inflammatory conditions that predispose to developing cancer. Intrinsic ones rely on genetic alteration, which is the reason for inflammation and cancer development. Both intrinsic and extrinsic pathways result in the activation of transcriptional factors such as Nuclear Factor-κB (NF-κB), STAT3, and hypoxia-inducible factor 1α (HIF1α) [46,47], as shown in Figure 1.

NFκB is a nuclear transcription factor involved in inflammation, carcinogenesis, proliferation, and apoptosis. It is also responsible for DNA damage by reactive oxygen species (ROS) secretion [17]. STAT3 is a transcription factor that stimulates apoptosis inhibitors and angiogenesis inducers. It is also involved in regulating the cell cycle, thus playing a significant role in tumor development and progression. Likewise, HIF1α is a transcriptional factor promoting cell proliferation and survival [48]. Beyond the influence of chronic inflammatory processes on cell proliferation, they also participate in the generation of ROS and reactive nitrogen species (RNS), epithelial-mesenchymal transition (EMT), angiogenesis, and metastasis [17]. Furthermore, COX-2-overexpressing cells produce large amounts of vascular endothelial growth factor (VEGF). VEGF is involved in angiogenesis, which is needed by cancer patients to provide large amounts of nutrients and also to promote metastasis [49].

Since chronic inflammation could be an important trigger for CRC development, targeting the eicosanoids pathway seems to be a good strategy for cancer prevention. There is strong data about aspirin’s chemopreventive effects in CRC [50,51]. Aspirin—acetylsalicylic acid—belongs to non-steroidal anti-inflammatory drugs, whose mechanism of action relies on the inhibition of cyclooxygenase 1 (COX1) and cyclooxygenase 2 (COX2). Early studies about aspirin’s chemopreventive action in CRC come from the 1990s [52]. Since then, many studies have been conducted that have proved that aspirin causes a reduction in CRC risk, but only after continuous and long-term use. Ma et al. [50] summarized these studies, prepared a meta-analysis of randomized controlled trial data, and showed that aspirin reduces the risk of mortality and the recurrence of CRC, which are significant problems in the struggle with CRC [53]. Unfortunately, this beneficial chemopreventive effect was also correlated with an increased risk of bleeding [2]. 

## 4. Chemopreventive Effect of Mesalazine on CRC

Recent studies suggest that, besides aspirin, long-term 5-ASA treatment reduces the risk of CRC developing in patients with IBD [54,55,56]. The first clinical findings about the chemopreventive role of 5-ASA come from 1994. Pinczowski et al. [54] showed that sulfasalazine significantly decreased the risk of CRC in patients with ulcerative colitis. Qiu et al. [36] conducted a systematic review with a meta-analysis to evaluate the effects of 5-ASA on the risk of colorectal cancer in patients with ulcerative colitis and Crohn’s disease. As a result, they showed that 5-ASA has a chemopreventive effect on CRC in IBD patients, with more benefits for UC patients than CD patients. To ensure effective treatment for reducing CRC risk in IBD patients, a mesalazine maintenance dosage of ≥1.2 g/day is needed. The European Crohn’s and Colitis Organization has recommended oral 5-ASA administration as chemoprevention in colitis-associated cancer (UC, CD).

Many in vitro studies show that mesalazine inhibits the growth and enhances apoptosis of CRC cell lines [37,38,39]. Moreover, studies conducted using animal models show that mesalazine inhibits tumor growth [57,58,59]. Notably, mesalazine treatment reduces the rate of proliferation of tumor cells without affecting the proliferation of normal epithelial cells [58,59]. Bus et al. [60] showed that topical administration of mesalazine for two weeks at a dosage of 4 g a day resulted in induced apoptosis in colorectal cancer cells, whereas it has no effect on the normal mucosa. 5-ASA’s chemopreventive mechanisms of action include interfering with the Wnt/β-catenin pathway, inhibition of cyclooxygenase and lipoxygenase mediators, antioxidant function, activation of the PPARγ pathway, and interfering with the EGFR pathway [61,62]. The chemical structure and mechanisms of action of mesalazine in CRC chemoprevention are shown in Figure 2.

## 5. Mesalazine’s Anticancer Mechanisms of Action

### 5.1. Mesalazine’s Effect on Cell Cycle

Experimental data indicate that 5-ASA reduces the survival and growth of CRC cells through the modulation of different replication checkpoints [63]. Reinacher-Schick et al. [64] showed that mesalazine inhibits the proliferation of colon cancer cells, probably through a specific accumulation of cells in mitosis. HT-29 cells treated with mesalazine accumulated in the G2/M phase. This mechanism differs from that of other NSAIDs (indomethacin and sulindac), which induce a robust G1. Additionally, they showed that mesalazine induced apoptosis in colon cancer cells, possibly through the activation of caspase-3. This examination elucidated that the effect of apoptosis was seen in no more than 10% of the tested sample, which is less than in other NSAIDs. However, Stolfi et al. [65] showed that mesalazine leads to the accumulation of CRC cells in the S phase. This effect was associated with the ubiquitination and proteasome-dependent degradation of CDC25A, which is known to regulate the G1/S transition and S phase progression.

In addition, mesalazine has a beneficial effect on replication fidelity. Gasche et al. [66] demonstrated that mesalazine improves replication fidelity independent of post-replication mismatch repair. They showed that mesalazine at the 5.0 mmol/L level reduced the mutation rate in the mismatch repair-deficient HCT 116 cell line. It proves that mesalazine can reduce the rate of mutation, which is correlated with a decreased speed of tumor progression.

### 5.2. Inhibition “Inflammogenesis of Cancer” and ROS Scavenging

Numerous studies indicate that chronic inflammation is a significant cancer risk factor [17,67]. Mesalazine is a weak COX and LOX inhibitor. Some existing data suggest that Mesalazine’s chemopreventive mechanism of action is due to its anti-inflammatory properties. Pharmacological inhibition of COX-2 can prevent CRC development, possibly by inducing apoptosis, reducing cell proliferation, or modulating angiogenesis [54,64]. Stolfi et al. [51] conducted interesting research using the HT-115 CRC cell line with functionally active COX-2 and the colorectal adenocarcinoma cell line with COX-deficient (DLD-1). In effect, treatment of HT-115 cells with mesalazine causes a reduction in prostaglandin E2 levels, which is correlated with decreased proliferation of these cells. Interestingly, DLD-1 cell lines treated with mesalazine also showed blocking of cell growth, but through different mechanisms of action [54,68]. Moreover, the inflammatory process often generates large amounts of ROS [69]. Numerous data indicate that 5-ASA acts as an oxygen-free-radical scavenger [70,71,72]. Oxidized 5-ASA regenerates using endogenous compounds like cysteine, glutathione, or ascorbic acid. Therefore, the drug is preserved and able to act even in tissues undergoing oxidative stress [73].

### 5.3. Mesalazine Affects the Wnt/β-Catenin Pathway

The 5-ASA antineoplastic mechanism of action in CRC is strongly related to suppressing the Wnt/β-catenin pathway [74]. Mutation of the suppressor gene adenomatous polyposis coli (APC) is a common cause of genetic changes leading to the development of colorectal cancer by nuclear accumulation of β-catenin. The product of the APC gene is a protein that is important in creating the destruction complex responsible for the regulation of the level of β-catenin. The degradation complex consists of axin, APC, casein kinase 1, and glycogen synthase kinase 3 (GSK3β). β-catenin is a protein that plays a key role in the Wnt/β-catenin pathway [75,76,77]. This pathway is important in regulating embryonic development, cell growth and differentiation, cell fate, and self-renewal of stem cells [78]. In the absence of a ligand, this complex is responsible for the phosphorylation of β-catenin. Then, phosphorylated β-catenin is ubiquitylated by ligase and degraded by the proteasome [75]. On the other hand, when a ligand is present, it binds to Frizzled receptors, which leads to the inactivation of GSK3β. Inactivated complex blocks β-catenin phosphorylation, allowing accumulation in the cytoplasm. β-catenin then translocates into the cell nucleus, where it activates the transcription factor TCF/LEF (T-cell factor/lymphocyte enhance factor), which increases cell proliferation by activating the transcription of target genes encoding proteins such as c-Myc, cyclin D1, and CD44 [75,79]. Mutations resulting in upregulated Wnt/β-catenin signaling have been demonstrated to be sufficient for early adenoma formation [74,80].

Recently, there have been many new reports on the effect of mesalazine on the Wnt/β-catenin pathway. Furthermore, these studies show that mesalazine interacts with this signaling pathway in different ways [81,82,83,84,85,86,87,88].

Bos et al. [81] showed that mesalazine inhibits the Wnt/β-catenin pathway via inhibition of protein phosphatase 2A (PP2A). Treatment with mesalazine leads to increased phosphorylation of PP2A, which is related to decreased PP2A enzymatic activity. This effect led to increased β-catenin phosphorylation and then degradation; thus, mesalazine treatment reduced the expression of Wnt/β-catenin target genes. Furthermore, 5-ASA can also act by inducing phosphorylation of β-catenin at threonine 41 and serine 45, leading in the same way to its degradation by the proteasome.

Parenti et al. [82] consider the involvement of μ-protocadherin in mediating the effect of 5-ASA on the Wnt/β-catenin pathway. Mucin and cadherin-like protein (MUCDHL), also called cadherin-related family member 5 (CDHR5), belong to the cadherin-related family, which showed the strongest stimulation in expression profiling of all cadherins after 5-ASA treatment of colon cancer cells. *MUCDHL* gene encoding μ-protocadherin is silenced during the carcinogenesis of CRC [83]. *MUCDHL* in CRC is up-regulated by mesalazine, which is further reported to confirm that mesalazine inhibits the Wnt/β-catenin signaling pathway. In a previous study, Parenti et al. [84] showed that induction of expression μ-protocadherin by mesalazine is regulated by the Kruppel-like factor 4 (KLF4)—transcription factor. KLF4 is able to inhibit Wnt signaling by preventing the interaction of β-catenin with TCF4. Additionally, researchers showed that mesalazine induces the expression of μ-protocadherin, leading to the sequestration of β-catenin on the plasmatic membrane [82].

Brown et al. [85] found another way of interacting 5-ASA with the Wnt/β-catenin pathway that relies on inhibiting epithelial phosphoinositide-3 kinase, which is related to enhanced expression of Wnt/β-catenin target genes. Further investigations showed that 5-ASA demonstrates antioxidant properties, which confirm increased levels of antioxidant catalase after 5-ASA treatment [85]. Additionally, 5-ASA decreased the inactive phosphatase and tensin homolog (PTEN), which is a major negative regulator of PI3K. H_2_O_2_ leads to the inactivation of *PTEN* expression, whereas the antioxidant properties of 5-ASA increase *PTEN* activity. This data suggests that the antioxidant properties of 5-ASA may be the predominant mechanism for 5-ASA chemoprevention.

In another study regarding the influence of mesalazine on Wnt signaling, Khare et al. [87] showed that administration of mesalazine in mice downregulates p21-activated kinase 1 (PAK1), a serine/threonine kinase required for the full activation of Wnt/β-catenin signaling.

Munding et al. [88] revealed the inhibitory effect of mesalazine on the Wnt/β-catenin pathway through in vivo observations. β-catenin/TCF signaling activity was significantly reduced, probably through decreased β-catenin levels. Moreover, 5-ASA also reduced β-catenin nuclear localization, changing the expression of β-catenin target genes. The effect of mesalazine on the Wnt/β-catenin pathway is shown in Figure 3.

### 5.4. Mesalazine Influences Homeostasis and Intestinal Healing

E-cadherin is the main mediator of cell-cell adhesion, which is important to maintain epithelial integrity. E-cadherin is a transmembrane molecule whose intracellular domain interacts with catenin, mainly β-catenin, which enhances adhesion properties. Data showed that disorders in maintaining the homeostasis of the intestinal epithelium are related both to the initiation and progression of CRC [83]. Impaired expression of E-cadherin is implicated in chronic gut inflammation and colorectal cancer [89]. In a study conducted by Losi et al. [83] on colon cancer cell lines, they demonstrated the effect of mesalazine on the expression level of µ-protocadherin, which interacts with the β-catenin-related pathway. The results suggest that an increase in the expression of this molecule may therefore have an inhibitory effect on the process of colon cancer carcinogenesis.

Furthermore, β-catenin forms a complex that maintains cell-cell adhesion by interacting with the intracellular domain of E-cadherin. Loss of E-cadherin leads to releasing β-catenin into the cytoplasm, from where it penetrates the cell nucleus, resulting in abnormal proliferation [75]. Khare et al. [87,89] showed that 5-ASA is the factor leading to increased membranous expression of E-cadherin and β-catenin. These findings testify about the mucosal healing possibilities of 5-ASA through enhanced cell-cell adhesion, which is often a disorder in both UC and colon cancer.

### 5.5. Mesalazine’s Effect on PPARγ Pathway and EGFR

Mesalazine is a ligand for peroxisome proliferator-activated receptor (PPAR) transcription factors, highly expressed in the colon, which play an important role in cell differentiation and proliferation [90,91]. An increase in cytoplasmic PPARγ and loss of nuclear one were observed in malignancy [86]. It has been shown that ligands capable of binding to PPARs and activating them, especially PPARγ and PPARα forms, influence the differentiation and induction of apoptosis of neoplastic cells. As a result, these ligands are potential drugs in anti-cancer and chemopreventive CRC therapy [92,93].

Rousseaux et al. [92] showed that 5-ASA increased PPARγ expression and promoted its translocation from the cytoplasm to the nucleus. Furthermore, PPARγ can retain β-catenin in the cytosol and reduce TCF transcriptional activity [93]. Activation of PPAR also enhances the proteasomal degradation of β-catenin [94].

Epidermal growth factor receptor (EGFR) is a protein whose activation leads to stimulating mitogenic and pro-survival signals [95]. It is strongly related to the pathogenesis of CRC and is also overexpressed in CRC. Mesalazine suppresses the activation of EGFR by inhibiting its phosphorylation. Mesalazine also leads to enhanced activity of the protein tyrosine phosphatases (PTPs), which negatively control EGFR activation [62].

### 5.6. Mesalazine Inhibits Proliferation of Colorectal Cancer Stem Cells

The theory of cancer stem cells (CSCs) describes that cancer cells are hierarchically organized. CSCs are a small subpopulation representing approximately 0.1–10% of all tumor cells that have the capacity to self-renew and differentiate into all types of cells in colon cancer. CSCs are involved in tumor initiation, maintenance, metastasis, and cancer recurrences [4,96,97,98]. Furthermore, these cells are responsible for chemotherapy and radiotherapy resistance. Many signaling pathways are dysregulated in CSCs; moreover, Hedgehog (Hh), Notch, and Wnt/β-catenin potentially regulate tumorigenesis in CSCs [99,100]. A recent study conducted by Dixon et al. [74] showed that 5-ASA suppresses β-catenin transcriptional activity, a key signaling pathway in stem cells. 5-ASA also inhibits the growth of adenoma cells and their stemness, which was demonstrated by repressing the expression of the stem cell marker LGR5 and protein CD133. Moreover, 5-ASA blocked the formation of adenoma-derived spheroids.

## 6. Conclusions

Over the last few decades, extensive research has been performed to indicate that mesalazine is effective in preventing CRC. The analyzed data showed that mesalazine has anti-neoplastic properties related both to its ability to inhibit COX enzymes and through COX-independent pathways. Importantly, these findings highlighted that the anticancer effect was not associated with changes in normal epithelial cells. Furthermore, mesalazine has an inhibitory effect on colon cancer stem cells, which are often involved in CRC recurrence. These observations, together with mesalazine’s mucosal healing properties, indicate that mesalazine could be a potential candidate to support patients after chemotherapy and reduce the number of cancer relapses. Further studies will be necessary to evaluate the potential effect on colon cancer stem cells and the possibilities of designing novel chemoprevention programs.

## Figures and Tables

**Figure 1 molecules-28-05081-f001:**
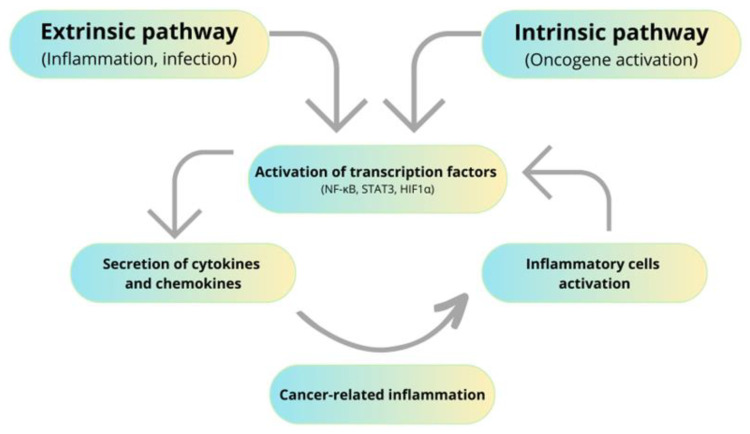
Influence of inflammation on cancer development.

**Figure 2 molecules-28-05081-f002:**
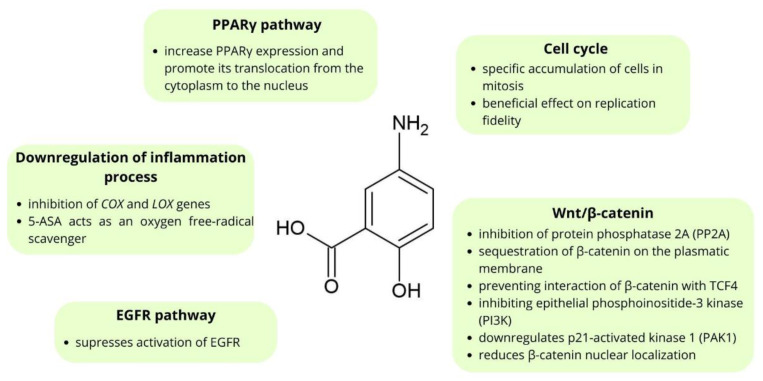
The main points of molecular chemoprevention by mesalazine.

**Figure 3 molecules-28-05081-f003:**
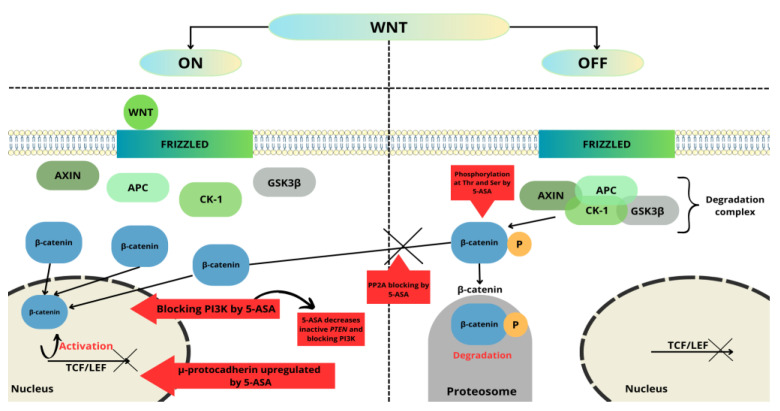
5-ASA’s major checkpoints in the Wnt/β-catenin pathway. This figure was created using the Servier Medical Art Commons Attribution 3.0 Unported Licence (http://smart.servier.com (accessed 4 May 2023)).

**Table 1 molecules-28-05081-t001:** Clinical trials with the use of mesalazine in the chemoprevention of diseases related to the large intestine.

Study Title	ClinicalTrials.gov Identifier	Phase	Status	Enrollment
Chemopreventive Action of Mesalazine on Colorectal Cancer: a Pilot Study for an “in Vivo” Evaluation of the Molecular Effects on β-catenin Signaling Pathway	NCT02077777	II	Completed	21
Mesalamine for Colorectal Cancer Prevention Program in Lynch Syndrome (MesaCAPP)	NCT03070574	II	Terminated	8
Mesalazine Effects in Sporadic Colorectal Adenoma Patients	NCT01894685	II	Completed	74
Mesalamine for Colorectal Cancer Prevention Program in Lynch Syndrome (MesaCAPP)	NCT04920149	II	Recruiting	260

## Data Availability

Available on request and with regulations.

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
