# Peer review of "Molecular Mechanisms of the Antitumor Effects of Mesalazine and Its Preventive Potential in Colorectal Cancer"

_molecules, 2023, doi:10.3390/molecules28135081_

Round 1

Reviewer 1 Report

After careful review, I have found that the manuscript is well-written. However, I have minor concerns that should be addressed before accepting the manuscript for publication.

1. The main concern relates to the topic of the manuscript. Most of the molecular mechanisms of Mesalazine have been studied in vitro, while the pathways studied in vivo have shown Mesalazine as a chemotherapeutic agent rather than a chemopreventive one, except for a few studies. Therefore, I suggest that the authors modify the title of the manuscript, as the term "chemopreventive" is misleading since the molecular mechanism of Mesalazine has mostly been studied in vitro.

2. Additionally, the authors claim that Mesalazine is free from side effects, but they only cite one study to support this claim.

Author Response

Review 1:

Comments:

Quality of English Language

( ) I am not qualified to assess the quality of English in this paper
( ) English very difficult to understand/incomprehensible
( ) Extensive editing of English language required
( ) Moderate editing of English language required
( ) Minor editing of English language required
(x) English language fine. No issues detected

Is the work a significant contribution to the field? 4/5

Is the work well organized and comprehensively described? 4/5

Is the work scientifically sound and not misleading? 4/5

Are there appropriate and adequate references to related and previous work? 4/5

Is the English used correct and readable? 4/5

Answer:

Our work has undergone minor modifications. We tried to take into account all the comments of the Reviewer increasing the quality of our work.

Comment:

“After careful review, I have found that the manuscript is well-written. However, I have minor concerns that should be addressed before accepting the manuscript for publication”.

Answer:

We thank the Reviewer for appreciating our work. We tried to refer to the Reviewer's comments and we hope that in its current form the work will be fully acceptable.

Comment:

“The main concern relates to the topic of the manuscript. Most of the molecular mechanisms of Mesalazine have been studied in vitro, while the pathways studied in vivo have shown Mesalazine as a chemotherapeutic agent rather than a chemopreventive one, except for a few studies. Therefore, I suggest that the authors modify the title of the manuscript, as the term "chemopreventive" is misleading since the molecular mechanism of Mesalazine has mostly been studied in vitro”.

Answer:

We would like to appreciate your accurate comment regarding the title of the manuscript. Therefore, we decided to change the title to: “Molecular Mechanisms of Antitumor Effects of Mesalazine and Its Preventive Potential in Colorectal Cancer”

Comment:

Additionally, the authors claim that Mesalazine is free from side effects, but they only cite one study to support this claim”. 

Answer:

We would like to thank the Reviewer for his attention to the literature. We include an additional link to the article: “Systematic review: the use of mesalazine in inflammatory bowel disease” by R. Bergman and M. Parkes, in which the authors note the minor side effects following the use of mesalazine. We have also modified the indicated sentence in the text.

Reviewer 2 Report

Dear Authors,

The review summarizing knowledge on the molecular mechanisms of mesalazine action with its chemopreventive effects is interesting and topical because of wide distribution of colorectal cancer in the world. The manuscript is well organized, clear and comprehensive. There are some lacks, in my opinion, of relevant references (please, see the attached PDF-file), several unnecessary abbreviations were used. These minor errors do not reduce the value of the review.

The quality of English is good with minor mistakes.

Author Response

Review 2:

Comments:

Quality of English Language

( ) I am not qualified to assess the quality of English in this paper
( ) English very difficult to understand/incomprehensible
( ) Extensive editing of English language required
( ) Moderate editing of English language required
(x) Minor editing of English language required
( ) English language fine. No issues detected

Is the work a significant contribution to the field? 4/5

Is the work well organized and comprehensively described? 4/5

Is the work scientifically sound and not misleading? 5/5

Are there appropriate and adequate references to related and previous work? 5/5

Is the English used correct and readable? 5/5

Answer:

We thank the Reviewer for the high evaluation of the manuscript. Our paper has undergone some minor revisions, which will improve the quality of the overall work.

Comment:

“The review summarizing knowledge on the molecular mechanisms of mesalazine action with its chemopreventive effects is interesting and topical because of wide distribution of colorectal cancer in the world. The manuscript is well organized, clear and comprehensive. There are some lacks, in my opinion, of relevant references (please, see the attached PDF-file), several unnecessary abbreviations were used. These minor errors do not reduce the value of the review”.

Answer:

We would like to thank the Reviewer for the valid comments. As recommended, we have made the suggested changes to the manuscript.

Comment:

Comments on the Quality of English Language

„The quality of English is good with minor mistakes”.

Answer:

We apologize for minor language errors and thank the Reviewer for pointing them out. We have made a change. Additional linguistic proofreading is also one of the steps in the procedure for publishing papers in MDPI, and therefore, in addition to our correction, there will be a linguistic evaluation at the publisher itself before the publication is made publicly available.

Reviewer 3 Report

This review manuscript summarizes the molecular mechanisms of mesalazine in Colorectal Cancer. Although it is not clear what is the molecular target of mesalazine in cells, authors have compiled the latest understanding of molecular basis of mesalazine in various signal transductions. The manuscript is well organized and worth to publishing in this journal.

A few minor points, I found some obscure terms in the manuscript. For example, at P8 lane 355, "5.6. Mesalazine inhibits colorectal cancer stem cells", I think it means mesalazine inhibits proliferation of cancer stem cells or stemness of the cells, i.e. differenciation. Please check the text and make it clear what authors are pointing out.

Author Response

Review 3:

Comments:

Quality of English Language

(x) I am not qualified to assess the quality of English in this paper
( ) English very difficult to understand/incomprehensible
( ) Extensive editing of English language required
( ) Moderate editing of English language required
( ) Minor editing of English language required
( ) English language fine. No issues detected

Is the work a significant contribution to the field? 4/5

Is the work well organized and comprehensively described? 5/5

Is the work scientifically sound and not misleading? 4/5

Are there appropriate and adequate references to related and previous work? 4/5

Is the English used correct and readable? 3/5

Answer:

Our work has undergone minor modifications. We tried to take into account all the comments of the Reviewer increasing the quality of our work. The manuscript has also been linguistically corrected again. Additional linguistic proofreading is also one of the stages of the procedure of publishing articles in MDPI, therefore, in addition to our proofreading, before publishing the publication, a linguistic assessment will be carried out by the publisher himself.

Comment:

“This review manuscript summarizes the molecular mechanisms of mesalazine in Colorectal Cancer. Although it is not clear what is the molecular target of mesalazine in cells, authors have compiled the latest understanding of molecular basis of mesalazine in various signal transductions. The manuscript is well organized and worth to publishing in this journal”.

Answer:

We would like to thank the Reviewer for his time, good opinion and recommendation of our manuscript for publication in the MDPI Molecules.

Comment:

“A few minor points, I found some obscure terms in the manuscript. For example, at P8 lane 355, "5.6. inhibits Mesalazine colorectal cancer stem cells", I think it means mesalazine inhibits proliferation of cancer stem cells or stemness of the cells, i.e. differenciation. Please check the text and make it clear what authors are pointing out”.

Answer:

We greatly appreciate the Reviewer noticing this mistake. By writing “Mesalazine inhibits colorectal cancer stem cells" we meant inhibition of the proliferation of CRC stem cells. We have corrected the paragraph as noted.